# Evaluating the Facilitators, Barriers, and Medical Outcomes Commensurate with the Use of Assistive Technology to Support People with Dementia: A Systematic Review Literature

**DOI:** 10.3390/healthcare8030278

**Published:** 2020-08-18

**Authors:** Clemens Scott Kruse, Joanna Fohn, Gilson Umunnakwe, Krupa Patel, Saloni Patel

**Affiliations:** School of Health Administration, Texas State University, San Marcos, TX 78666, USA; joannafohn@gmail.com (J.F.); gumunnakwe@gmail.com (G.U.); krupapatel94.kp6@gmail.com (K.P.); p1995saloni@gmail.com (S.P.)

**Keywords:** dementia, assistive technology, caregiver, cognitive disorder, stress

## Abstract

*Background*: Assistive technologies (AT) have been used to improve the daily living conditions of people living with dementia (PWD). Research supports the positive impact of the use of AT such as decreased burden on caregivers and behavioral support for people with dementia. Four reviews in the last six years have analyzed AT and PWD, but none have incorporated the dimension of medical outcomes. *Objectives:* The purpose of this review is to identify the facilitators, barriers, and medical outcomes commensurate with the use of AT with PWD. *Method:* This review queried The Cumulative Index of Nursing and Allied Health Literature (CINAHL), Web of Science, Science Direct, and PubMed databases for peer-reviewed publications in the last five years for facilitators, barriers, and medical outcomes commensurate with the use of AT with PWD. The study is reported and conducted in accordance with the Preferred Reporting Items for Systematic Reviews and Meta-Analysis (PRISMA) and the Kruse Protocol for conducting a systematic review. *Results:* 48 studies were analyzed. Fourteen types of AT, 17 facilitators, 17 barriers, and 16 medical outcomes were identified in the literature. The two most frequently mentioned ATs were cognitive stimulators (9/48, 19%) and social robots (5/48, 10%). The two most frequently mentioned facilitators were caregivers want AT (8/68, 12%) and enables increased independence (7/68, 10%). The top two barriers were cost (8/75, 11%) and PWD reject AT (8/75, 11%). The top medical outcomes were improved cognitive abilities (6/69, 9%), increased activities of daily living (ADLs), and increased autonomy (each at 5/69, 7%): Zero negative outcomes were reported. *Conclusion:* The systematic review revealed the positive relations that occur when PWD and their caregivers use AT. Although many reservations surrounding the use of AT exist, a majority of the literature shows a positive effect of its use. Research supports a strong support for AT by caregivers due to many positive medical outcomes, but also a reticence to adopt by PWD. If ATs for PWD are a way to reduce stress on caregivers, barriers of cost and complexity need to be addressed through health policy or grants.

## 1. Introduction

### 1.1. Rationale

Dementia describes a group of symptoms affecting a person’s cognitive abilities severely enough to interfere with their daily life [1,2]. Currently, over 46 million people live with dementia, and the numbers are expected to increase with the aging of society to 131.5 million by 2025. Dementia is a condition affecting older people and has an impact on families, care givers and society. Commonly, family members are seen to be the care givers to support people with dementia and are usually untrained for this demanding role. The situation, of increasing numbers of those who suffer from dementia combined with an untrained social network to care for them, is rife for a technological intervention. 

The etiology and disease stage of dementia patients can be characterized based on their cognitive, behavioral, motor, and functional symptoms [3]. Research supports the incorporation of technology with dementia patients because it plays a role in preventing their cognitive and physical decline. Technological intervention could further propose solutions to the challenges and barriers that limit care in dementia patients [4].

Various technological interventions have been introduced for functions such as of memory support, safety and security for the patient, training for both the caregiver and patient, care delivery, overall treatment, and social interaction [5]. However, barriers to technological treatment, such as privacy concerns, adaptation, and design choice, could result in a decline of the technology’s effectiveness [6]. Assistive technologies for dementia patients vary in their design: for the occasional use of the patient, by the patient, or on the patient (e.g., caregivers) [7]. Technological interventions for dementia patients could provide quality care measures to accommodate daily needs.

Four reviews in the last six years were published. They recognize the rapidly developing world of intelligent assistive technologies (ATs), and attempted to compile comprehensive lists of such devices and their benefit [8,9,10,11]. Date ranges spanned 5–16 years, and the reviewers included studies of both dementia patients and their carers. However, none of these reviews identified the medical outcomes commensurate with the use of AT, which is a basic requirement for a systematic review [12].

### 1.2. Objectives

The purpose of this review is to compile a current and comprehensive list of facilitators and barriers to the adoption of, and medical outcomes commensurate with, the use of AT by PWD and their carers to perform activities of daily living independently. The results of this review should enable future studies to explore the modifications required for AT to support people with dementia while providing ease of use for the care givers.

## 2. Methods

### 2.1. Protocol and Registration

This review followed the Kruse protocol for conducting a systematic review published in 2019 [13], reported in accordance with Preferred Reporting Items for Systematic Reviews and Meta-Analysis (PRISMA) [14]. This review was registered with PROSPERO on 7 May 2020: PROSPERO ID CRD42020182167. In accordance PROSPERO rules, the registration was completed before analysis began.

### 2.2. Eligibility Criteria

Studies were eligible for this review if participants were diagnosed with dementia or they were care givers (carers) of those with dementia (regardless of stage, gender, race, ethnicity, or age), if the intervention was AT, if they were published in a quality, peer-reviewed journal in the English language in the last five years, and it they report either facilitators to adoption, barriers to adoption, or medical outcomes associated with the use of AT. Five years was chosen due to the rapidly advancing field of AT development. A quality assessment will be performed on each article using the Johns Hopkins Nursing Evidence-Based Practice Rating Scale (JHNEBP) [15]. Any study below IV C will be discarded.

### 2.3. Information Sources

Reviewers queried four databases: The Cumulative Index of Nursing and Allied Health Literature (CINAHL), PubMed (MEDLINE), Web of Science, and Embase (Science Direct). Databases were filtered for the last five years. Database searches occurred between 1–15 February 2020. We expected to find advances in AT and greater level of adoption of AT due to both availability of these devices and the growing number of PWD. We hoped to find fewer barriers to the adoption of AT for PWD.

### 2.4. Search

Reviewers initially conducted a search on Google Scholar using general terms. When 10 articles were found on the subject, reviewers collected the key terms from these studies to help form a Boolean search string. Using the PubMed Medical Subject Headings (MeSH), reviewers used the terms gathered from the 10 articles to examine how they were indexed and categorized. Once a Boolean search string was assembled, it was tested out several times in PubMed and customized for maximum, most effective yield. The final search string was (“self-help devices” OR “selfhelp devices” OR “self help devices” OR “assistive technology” OR “telemonitoring” OR “tele-monitoring” telemedicine) AND (dementia OR “cognitive impairment”). We used this same search string for all databases. Reviewers filtered out other reviews and helped the database focus on academic or peer-reviewed journals over the last five years.

### 2.5. Study Selection

Following the Kruse protocol, reviewers conducted three consensus meetings [13]. The first was to select the group of studies to analyze. Once the search identified a large group to screen, the literature matrix manager downloaded the article details from each research database into an Excel literature matrix which served as the applied form from which to extract data. The group leader assigned workload to the group to ensure all abstracts would be screened by at least two reviewers. Reviewers read their assigned abstracts and screened them against the objective statement making a keep or discard recommendation on the shared spreadsheet. Once all abstracts were screened, the group met to discuss disagreements in recommendations. A final determination was made by the end of the meeting by the group leader, by asking another member of the group to read the abstract. A kappa statistic was calculated based on this process [16].

### 2.6. Data Collection Process

Once the final group of articles for analysis was identified, the group leader assigned workload to ensure all articles were analyzed by at least two reviewers. Reviewers used the applied form as the data extraction tool to collect Participants, Intervention, Comparison, Outcome, Study design (PICOS) and make general observations. Data items on this spreadsheet were published in the protocol [13]. Reviewers independently analyzed articles, extracting all standardized data items as well as general observations commensurate with the objective statement [13]. At consensus meeting 2, these observations were shared, and a narrative analysis was conducted [17]. The narrative analysis attempts to make sense of the observations. From the observations, common threads were identified. Being mindful of the common threads, or themes, reviewers carefully read their articles another time to flush out additional occurrences. Using the themes, reviewers examined interactions between facilitators, barriers, and medical outcomes to determine if some interventions were more consistently successful or problematic than others.

### 2.7. Data Items

The applied form collected the following data items: participants, AT intervention, study design, results compared to a control group (where applicable), facilitators and barriers to the use of AT, medical outcomes, sample size, bias within studies, effect size, country of origin, statistics used, a quality assessment from the JHNEBP, and general observations about the article that would help interpret the results [15].

### 2.8. Risk of Bias within and across Studies

General observations of bias are collected throughout the analysis phase. Bias is discussed in the second consensus meeting along with other observations. Key observations of bias, such as selection bias, are discussed because these could limit the external validity of the results. The JHNEBP was used to assess the risk and quality of each study analyzed. The cumulative evidence from the analysis (selective reporting, etc.) is discussed at the second consensus meeting. The JHNEBP is comprised of five levels for strength of evidence and three levels for quality of evidence. The strength of evidence for level 1 is an experimental study or RCT. Level 2 is strictly for quasi-experimental studies. Level 3 is for non-experimental, qualitative, or meta-synthesis studies. Level 4 is for opinion of nationally recognized experts based on research evidence or consensus panels. Level 5 is for opinions of experts that is not based on research evidence. The quality of evidence is listed as A (high), B (good), or C (low quality or major flaws). Each of these levels contain specifics for research, summative reviews, organizational, and expert opinion. For instance, research in level A must have consistent results with sufficient sample size, adequate control, and definitive conclusions. Research in level B must have reasonably consistent results, sufficient sample size, some control, and definitive conclusions. Research at level C has little evidence with inconsistent results, insufficient sample size, and conclusions that cannot be drawn from the data. Articles with a strength of evidence rating below Level 4 will be screened out. Quality of evidence below level B are highly suspect and must have full consensus of the group to be kept for analysis.

### 2.9. Summary Measures

The review analyzed studies with qualitative, quantitative, and mixed methods, so the summary measures sought were not consistent. The preferred summary statistic would be the risk ratio, but descriptive statistics and means’ comparisons (student-*t*) are also sufficient. Summary statistics were discussed at the second consensus meeting.

### 2.10. Synthesis of Results

This subsection addresses a meta-analysis. This is a systematic review. This section is provided to assure reviewers that PRISMA had been followed. It will be deleted prior to publication, if accepted.

### 2.11. Additional Analysis

At the second consensus meeting, a narrative analysis will be performed to group observations into themes. These themes will be measured across all articles analyzed and reported in summary statistics in a series of affinity matrices. The narrative analysis summarized themes for facilitators, barriers, and medical outcomes. These will be reported in affinity matrices.

## 3. Results

### 3.1. Study Selection

The study selection process performed is illustrated in Figure 1. A kappa statistic was calculated after the first consensus meeting (k = 0.85), which indicates strong agreement [16,18]. After screening, removing duplicates, and assessing for eligibility, the 48 articles chosen for analysis came from CINAHL (16, 33%), Web of Science (16, 33%), PubMed (15, 31%), and Science Direct (1, 2%).

### 3.2. Study Characteristics

Using the applied form, reviewers collected several standard items used for summary, such as PICOS. A PICOS table is provided in Table 1. Additional items were collected for analysis, such as forms of AT interventions, facilitators, and barriers to the use of assistive technologies, and the medical outcomes observed from those older adults using AT solutions [13]. These are presented in Table 2. Table 1 and Table 2 lists articles in reverse chronological order: 2019 (4) [3,19,20,21], 2018 (12) [22,23,24,25,26,27,28,29,30,31,32,33], 2017 (14) [34,35,36,37,38,39,40,41,42,43,44,45,46,47], 2016 (11) [48,49,50,51,52,53,54,55,56,57,58], 2015 (7) [59,60,61,62,63,64,65].

### 3.3. Risk of Bias within Studies

Reviewers recorded observations of bias at the study level. The most common form of bias was selection bias. Examples of selection bias were all: participants had experience with technology [23], same site [25,28,30,34], or a disproportionately large sample that was male [22,30,36]. These examples of bias limit the external validity of the results.

### 3.4. Results of Individual Studies

Reviewers collected their observations of intervention and medical outcomes during the analysis phase. The narrative analysis of their observations identified themes. A summary of these themes is listed in Table 1. Repetition in the frame of a theme is due to multiple observations from the same article for that theme. For instance, the theme *increased talking* comprised observations of “increased utterances” and “increased sustained conversations” [19]. A translation from observations to themes for interventions, medical outcomes, facilitators, and barriers us listed in Table A1, Table A2 and Table A3. Reviewers collected the standard PICOS fields and included them in Table 1. Additional data collected is displayed in Table A4 and Table A5: bias, statistics, country of origin, and quality assessments.

### 3.5. Synthesis of Results

This subsection addresses meta-analyses. This is a systematic review. This section will be deleted after the review process. It is included to reassure reviewers that we followed the PRISMA checklist.

### 3.6. Risk of Bias across Studies

Table 3 summarizes the quality indicators identified by the JHNEBP tool [15]. The most prevalent assessment in the strength of evidence (panel a) was level III, followed by I, II, and IV. For quality of evidence (panel b), the most frequently assessed level was level B, followed by A. It is certainly preferable for the strength of evidence to be closer to level I, but that has not resulted from the screening and selection process. This limitation will be addressed later.

### 3.7. Additional Analysis

#### 3.7.1. Interventions of AT

Consensus meeting three identified eight themes and six individual observations that corresponded with AT for PWD. These are listed in Table 4. In the interest of brevity, only the first 60% will be listed. The intervention most often noted was an interview [3,22,23,24,25,29,31,39,43,44,49,57,59,61,63,65]. Researchers interviewed users of AT (PWD and carers) in 16/48 studies (33%). Three of these studies originated in the United Kingdom, two from Norway, two from the Netherlands, and the rest were from single originations: Greece, United States, Pakistan, Finland, Australia, United Kingdom/Italy/Malaysia, Canada, Sweden, and Taiwan. The theme cognitive stimulation (natural language presentation through a Wizard-of-Oz presentation, intelligent cognitive assistant, computerized help information and interaction project, wearable cameras, and evidence-based reasoning and problem solving training) occurred in 9/48 studies (19%) [26,36,37,51,53,54,58,60,65]. Four of these studies originated from the United Kingdom, while the rest were from single originations: France, Scotland, Saudi Arabia, Canada, and Netherlands/Germany/Belgium. Socially assistive robots (artificial intelligence system designed to interact with humans and other robots) were identified in 5/48 studies (10%) [19,35,42,47,62]. These studies all came from single originations: Mexico, Italy/Ireland, Canada, New Zealand, and the United States.

#### 3.7.2. Facilitators to the Adoption of AT

Fourteen themes and three individual observations were identified as facilitators to the adoption of AT for PWD. These are listed in Table 5. In the interest of brevity, only the top 40% most frequently observed will be reported (other than those not reported). The theme *caregivers want* AT (AT more ethical than physical barriers, users are open-minded to robotic assistance and sensors, usefulness is high, AT enables PWD to live at home more safely) occurred in 8/68 occurrences (12%) [36,38,39,43,44,47,53,63]. These studies all came from single originations: Australia, Scotland, Canada, United Kingdom, United Kingdom/Italy/Malaysia, Spain, Netherlands, Germany/Belgium, and Taiwan. The theme *increased independence* (keeps PWD home longer, helps structure everyday events) occurred in 7/68 occurrences (10%) [3,30,40,46,51,53,59]. Three of these studies originated from Sweden, and two were from Canada, while the rest were from single originations: Netherlands, Netherlands/Germany/Belgium. The theme *increased safety* (increases a sense of security, increases safety) occurred in 6/68 occurrences (9%) [3,25,30,35,41,57]. These studies all came from single originations: Netherlands, Sweden, Norway, United States, New Zealand, and United Kingdom.

#### 3.7.3. Barriers to the Adoption of AT

Eleven themes and six individual observations were identified as barriers to the adoption of AT for PWD. These are listed in Table 6. In the interest of brevity, only the first 50% of the most frequently observed occurrences will be reported (other than those not reported). The theme of cost was identified in 8/75 occurrences (11%) [3,20,35,39,40,52,61]. Two of these studies came from Canada while the rest were from single originations: Netherlands, Australia, New Zealand, Switzerland, and the United Kingdom. The theme PWD *do not want AT* (PWD reject AT, users do not want a robot, users immediately returned the device, devices are unfamiliar, and participants feel targeted and embarrassed) occurred in 8/75 occurrences (11%) [22,23,24,47,55,62]. Three of these studies originated from the United States, while the others were from single originations: Greece, Canada, Pakistan, Taiwan, and the United Kingdom. The theme *complex interfaces* (pushing buttons does not always yield the desired result, system was perceived to be complex, complex setup) occurred in 7/75 occurrences (9%) [3,23,25,29,30,51,53]. These studies all came from single originations: Netherlands, Greece, Sweden, Finland, Norway, Canada, Netherlands/Germany/Belgium. The theme *development must improve capabilities* (false alarms, would only be worn for a few hours due to weight, technical problems, difficult to understand) occurred in 7/75 occurrences (9%) [25,26,29,42,43,48,53]. The theme *degenerative nature of dementia complicates the timing of* AT used (timing is important – before it is too late, dementia’s progressive nature makes useless today AT that may have been helpful only a short while ago, client’s reduced cognitive and physical abilities greatly reduces the effectiveness of AT more each day) occurred in 7/75 occurrences (9%) [24,25,36,40,42,45,52]. These studies occurred in single originations: Norway, United States, Italy/Ireland, Canada, Scotland, Netherlands, and Switzerland.

#### 3.7.4. Medical Outcomes Commensurate with the Adoption of AT

Twelve themes and four individual observations of medical outcomes were recorded commensurate with the adoption of AT. These are listed in Table 7. In the interest of brevity, only the first 40% will be reported. The theme of *improved cognitive abilities* (reduced sensory loss, increased global cognition, decreased cognitive decline) occurred in 6/69 occurrences (9%) [3,34,56,58,59,60]. Two of these studies originated from the United Kingdom, while the others came from the Netherlands, Australia, Greece, and Sweden. The theme of *increased* ADLs occurred in 5/69 occurrences (7%) [27,32,44,56,60]. Two of these studies originated from the United Kingdom, while the others came from the United States, Spain, and Greece. The theme of *increased autonomy* occurred in 5/69 occurrences (7%) [23,25,26,30,59]. Two of these studies originated from Sweden, while the others came from Greece, Norway, and France. The theme of *improved memory* (helps patients recall past events) occurred in 4/69 occurrences (6%) [34,48,65]. These studies originated from Australia, Sweden, and the United Kingdom. The theme of *improved overall health* (improved emotional health, improved psychological wellbeing, improved hearing, better tracking of health monitoring through AT) occurred in 4/69 occurrences (6%) [22,33,35,55]. These studies originated from Germany, Pakistan, New Zealand, and the United States. The theme of *improved mood* (improved emotional wellbeing, improved negative emotions, enabled greater enjoyment) occurred in 4/69 occurrences (6%) [33,34,41,64]. These studies originated from Germany, Australia, British Columbia, and the United States.

### 3.8. Interactions between Observations

The intervention of eHealth resulted in two instances of increased communication [45,64]. The intervention of cognitive stimulation resulted in two instances of increased independence [51,53]. When researchers interviewed users of AT, there were two instances of PWD do not want AT [22,63] and two instances of increased autonomy [23,25]. The intervention of social robots resulted in two instances of PWD do not want AT [47,62].

## 4. Discussion

### 4.1. Summary of Evidence

Fourteen interventions of AT in 48 studies revealed 17 facilitators, 17 barriers, and 16 positive or neutral medical outcomes. The two most frequently mentioned AT were cognitive stimulators (9/48, 19%) and social robots (5/48, 10%). The two most frequently mentioned facilitators were caregivers want AT (8/68, 12%) and enables increased independence (7/68, 10%). The top two barriers were *cost* (8/75, 11%) and PWD do not want AT (8/75, 11%). The top medical outcomes were improved cognitive abilities (6/69, 9%), increased ADLs, and increased autonomy (each at 5/69, 7%): Zero negative outcomes were reported. Interactions of note occurred with interventions of eHealth and cognitive stimulation, which resulted in multiple observations of increased communication and increased independence, respectively.

Two interesting dichotomies were observed. One study noted no effect in depression, while others noted improved depression [33,41], behavior [24,33,56], mood [33,34,41,64], coping [3,33,62], and decreased stress [25,48]. Also, caregivers want AT [36,38,39,43,44,47,53,63], while PWD do not want AT [22,23,24,47,55,62]. Caregivers feel a sense of relief knowing sensors are in place, and they feel these sensors are more ethical than using physical barriers to prevent elopement, but the complex and foreign nature of the sensors coupled with false alarms cause PWD to reject the same devices that increase their sense of autonomy [23,25,26,30,59].

The observations that AT needs additional development to improve interfaces, reduce false alarms, and improve functionality of devices was no surprise. The challenging aspect to development is seen in the observation that the degenerative nature of AD complicates the timing of AT used. Even the ideal interface, that has been proven effective for a PWD for months, may be obsolete one specific morning due to the degenerative nature of AD, because the PWD may forget how to use it [3,23,52]. Developers might consider AT with phases of simplicity of operation (which might result in reduced capabilities) so that the AT can advance along with the PWD.

Health policy makers should consider augmenting AT development through grants or subsidies. As the aging of society progresses, so will the prevalence of PWD [2]. Development of AT for PWD should be a priority. This corresponds with the theme needs more government involvement.

Social services workers should keep a list of ATs available to PWD at different stages of AD. Caregivers, often spouses or close family, are already experiencing high levels of stress and concern. A general lack of awareness was observed in one study [39]. The combination of more government involvement and funded development should enable such lists to be readily available and regularly updated.

Caregivers should know that AT exists for PWD, and that they are not alone in the struggle to care for their loved ones while also maintaining their own life. AT currently exists, and it improves the quality of life for both PWD and the caregivers [27,32,44,56,60]. The research strongly supports a desire by caregivers for AT. Although a barrier was observed of complex interfaces, several other studies noted the existence AT with simple interfaces with visual and auditory reminders and constant training to control for forgetfulness. Some AT can be provided by the government as part of national health plans.

The findings of this review are commensurate with the other four reviews from the previous six years [8,9,10,11]. The AT reviewed is similar to each of these articles [8]. The decline in memory and cognition is common [9]. One semantic difference is that one review called AT intelligent ATs, and this same review provides an extensive index of AT for clinicians and other stakeholders involved in the management of PWD [10].

The findings in this review are also commensurate with four others from the last year, and the AT reviewed are similar [66,67,68,69,70]. AT interventions can be beneficial to quality of life, social interaction, reducing neuropsychiatric symptoms such as depression, anxiety, and agitation, but continued use of AT becomes problematic due to the progressive nature of dementia [69,70]. Both carers and PWD shared observations on aspects of AT such as ease of use, stability and flexibility of technology, and the importance of privacy and confidentiality [67,68]. One finding that was not in this review was an improvement in pain management, as was found in a review from 2019 [69].

Future research in this area should examine the ethical concerns surrounding AT. If AT yields so many positive medical outcomes, and it is strongly desired by caregivers, why is it also rejected by PWD? Is AT perceived as an annoyance or a bother? Do some view AT as a constraint on liberty? The literature supports the use of AT. It is important for the subject of AT to perceive its value and willingly accept it.

### 4.2. Limitations

The researchers reviewed papers published between 2015 with 2020 and did not include studies outside this period. One limitation is publication bias for five years. Including some grey literature could have controlled for this limitation. We limited our review to four databases, CINAHL, PubMed, Web of Science, and Science Direct. The intent of including four databases was to mitigate the risk of selection bias. The limitation is that we did not include other databases that might have yielded other articles for consideration.

A team of reviewers determined the articles to be included in the study. This was also done to mitigate the risk of selection bias. The risk of this practice, however, is that the team may have differed in their selection processes. To mitigate this risk, researchers held consensus meetings, identified the research objective, and received multiple reviews for each article. The limitation is that there may not have been enough consensus meetings. The kappa statistic shows the consensus meetings were effective, but a stronger level of agreement is possible.

The assessment of both strength of evidence and quality of evidence was not as high as preferred. The most common strength of evidence was III, while the preference would have been for more level I. To control for this limitation in the future, additional years could have been considered so that more experimental studies or RCTs could have been included in the analysis.

## 5. Conclusions

The research supports the use of AT with PWD, and many positive or neutral medical outcomes are associated with this practice. This review presents facilitators and barriers associated with the use of AT by PWD, and it presents medical outcomes commensurate with this intervention. It is consistent with other reviews on the same or similar topics. Policy makers and carers should focus on the enablers listed in this review and work to eliminate the barriers to the adoption of AT. Such practice would help policy makers enable additional PWDs and carers to experience the positive aspects of this intervention, for as long as the progressive nature of dementia will allow. Particular focus should be given to helping PWD accept AT based on the positive aspects of their use.

## Figures and Tables

**Figure 1 healthcare-08-00278-f001:**
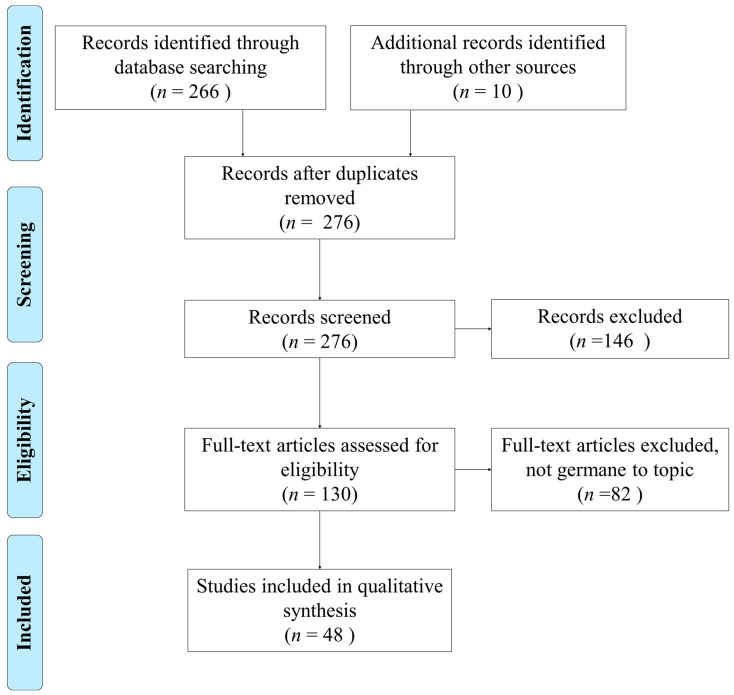
Preferred Reporting Items for Systematic Reviews and Meta-Analysis (PRISMA) flow diagram of the literature search and selection process.

**Table 1 healthcare-08-00278-t001:** PICOS table.

Authors	Participants	Intervention	Results (Compared to Control Group)	Medical Outcomes Reported	Study Design
Cruz-Sandoval D. and Favela J. [19]	Older adults with dementia, ≥70, good diction and hearing	Socially assistive robots	No control group. The number of utterances increased when conversational strategies were used and PWD engaged in more sustained conversations. Participants enjoyed conversing with robots.	Increased number of utterances and sustained conversations	Single-subject research design (simple observational, without control group)
Thomas N.W.D., et al. [21]	One older adult with AD and his caregiver, ≥70	Telehealth home-based monitoring Ecological Valid, Ambient, Longitudinal and Unbiased Assessment of Treatment Efficacy in Alzheimer’s Disease (EVALUATE-AD)	No control group. Very little effect on depression, but the intervention may have stabilized the burden of care.	Zero effect on depression	Observational case study
Kenigsberg P.A., et al. [3]	Experts, age, race, and gender not reported	Several ATs were explored: smart calendar clocks, social robots, auto lighting, computer, smartphone	No control group. Many ATs show positive health outcomes and increase safety and wellbeing of users and caregivers, but cost is a barrier to adoption	Reduced sensory loss, Reduced isolation, Improved cognitive abilities (memory, judgment, decision making),	Delphi study and survey
Czarnuch S., et al. [20]	Caregivers, 79% female, race not reported	Predictive model to predict survey results. Survey instrument addressed ATs and activities of daily living (ADLs).	No control group. The predictive model predicted 84% of responses (20 daily tasks × 5 response levels per task).	Not reported	Mixed methods
Billis A., et al. [23]	Older adults, 80% female	Interviews about ATs	No control group. Elderly participants exploit technology for several daily tasks such as checking bus schedules, watching online videos, and communicating with friends	Increase in autonomy	Qualitative
Lancioni G.E., et al. [27]	Study 1: 8 participants with mild to moderate stages of disease. Study 2: 9 participants with lower half of the moderate or severe level of Alzheimer’s disease	Study 1—wireless Bluetooth earpiece linked to the tablet or smartphone. Study 2—walker with technology that provided stimulations and prompts	Patients started carrying out activities independently and accurately. They increased their ambulation levels and showed signs of positive involvement	Increase in ADLs	Quasi-experimental
Lancioni G.E., et al. [28]	Older adults with Alzheimer’s Disease	Use of walker with AT	No control group. Participants showed an increase in indices of positive involvement during intervention sessions	Increase in step frequency	Qualitative Pilot Study
Olsson A., et al. [30]	Older adults with Alzheimer’s Disease	Sensor technology with individually prerecorded voice reminders as memory support	No control group. Sensors increased a sense of security, independence, and self confidence	Increase in autonomy, Increased self confidence	Qualitative
Nauha L., et al. [29]	Eight nurses from assisted living facility taking care of five test subjects, three female family caregivers in a real home environment in care of their husbands, one female test subject living alone, and several nurses took part in her care during day.	Supportive devices and alarm systems	Large variations in usefulness and usability of different AT. For example, a safety bracelet was very useful and easy to use compared to medical dispenser or a smart flower stand	Not reported	Experiment
Holthe T., et al. [25]	Adults with young-onset dementia (YOD) and their care givers	Interview to discuss experiences with AT	No control group. AT can be both a relief and a burden.	Increase in autonomy, Decreased stress	Qualitative
Wilz G., et al. [33]	Caregivers (80.6% female), and those with moderate and severe dementia (51.3% female), race not reported	Telephone therapy for caregivers	Improvement was observed across several domains	Improved emotional wellbeing, Fewer symptoms of depression, Fewer physical health symptoms, Improved coping, Improved behavior	RCT
Valdivieso B., et al. [32]	Elderly high-risk patients with multiple chronic conditions	Telephone-based telehealth which adds technology for remote self-management	Quality of life was significantly higher for the intervention group	Improved quality of life indicators	Quasi-experimental
Collins, M. [24]	Occupational therapy practitioners that use ATs on clients with Alzheimer’s disease and related dementia	Interview	No control group. When using AT, occupational therapy practitioners must consider the stage of dementia their client is in, as well as the client’s performance skills. Assistive technology is primarily used with the Alzheimer’s disease and related dementia (ADRD) population to address safety concerns such as elopement, fall prevention, kitchen safety, and medical management.	Fewer medication errors, Fewer falls, elopement, and fewer kitchen safety incidents	Qualitative phenomenological approach
Asghar I., et al. [22]	People with dementia between the ages of 63 and 72 (mean = 68)	Interview	No control group. The AT producers should make user interface simpler and tailor future ATs to the specific requirements of the PWD	The use of technology in combination with balanced human care can yield better results for the wellbeing of the PWD; some participants used ATs for health monitoring purposes. These devices helped them to monitor their physical conditions like blood pressure, heartbeat rate, diabetes level, etc., on regular basis. This continuous health monitoring helped them to adapt a healthier lifestyle	Qualitative
Jouvelot P., et al. [26]	Older adults, 79% female	Lovely User Interface for Servicing Elders	No control group. 13/14 were able to interact with the intervention	Increased autonomy	Non-experimental
Thoma-Lürken T., et al. [31]	Formal and informal caregivers and experts in the field of AT	Interview to gain insight into the most important practical problems preventing people with dementia from living at home	No control group. Problems within three domains were identified: informal caregiver/ social network-related problems, high load of care, safety-related problems, decreased self-reliance	Not reported	Qualitative study using six focus group interviews
Kouroupetroglou C., et al. [42]	People with dementia	MARIO robot	No control group. PWD who engaged in the above testing phase were accepting of MARIO and liked his appearance but the multimodal interaction combining verbal and visual cues was, in some cases, challenging	Not reported	Questionnaire (qualitative and quantitative)
Jiancaro T., et al. [40]	Developers (including technical, clinical, and other specialists) working in the design and/or evaluation of technologies for people with AD or related dementias	Exploratory survey with developers	No control group. To summarize, developers are encouraged to consider the following design recommendations for their own projects: Acknowledge the considerable degree of diversity amongst users with dementia, potentially adopting user centered design (USD) techniques, such as designing for ‘‘extraordinary users’’; Recognize the important theoretical role that clinical specialists can fulfil concerning the use of design schemas; Stipulate precise usability criteria; Consider ‘‘learnability’’ and ‘‘self-confidence’’ as technology adoption criteria; Consider possible cost/adaptability tradeoffs during the design process; Encourage interdisciplinary communication via education and team engagement.	Not reported	Case-based survey (qualitative)
Jarvis F., et al. [39]	Occupational therapists	Survey on occupational therapists use and awareness of AT for PWD.	No control group. There is a limited understanding from occupational therapists about available interventions for PWD.	Not reported	Survey
Dethlefs N., et al. [36]	Elderly adults, 74% male, 10 with dementia	Natural language presentation (Wizard of Oz interface) of cognitive stimulation	No control group. Of the sorting activity, name-recall activity, quiz activity, and proverb activity, 83% reported enjoyment.	Not reported	Pilot study
Wang R.H., et al. [47]	Older adults and caregivers as dyads	Robots to assist in daily activities	Robots were well received by carers but resisted by PWDs.	Not reported	Qualitative analysis,
Toots A., et al. [46]	People with dementia	Participants were randomized to the high-intensity functional exercise program or a seated attention control activity	In people with dementia living in nursing homes, exercise had positive effects on gait when tested unsupported compared with when walking aids or minimum support was used.	Improved gait	Cluster-randomized controlled trial
Bahar-Fuchs A., et al. [34]	45 older adults, >65, with mild cognitive impairment (*n* = 9), mood-related neuropsychiatric symptoms (*n* = 11) or both (*n* = 25)	eHealth Tailored and adaptive computer cognitive training in older adults at risk for dementia	Participants in both conditions reported greater satisfaction with their everyday memory following intervention and at follow-up. However, participants in the computerized cognitive training (CCT) condition showed greater improvement on composite measures of memory, learning, and global cognition at follow-up. Participants with Mood-related Neuropsychiatric Symptoms (MrNPS) in the CCT condition were also found to have improved mood at three-month follow-up and reported using fewer memory strategies at the post-intervention and follow-up assessments. There was no evidence that participants with MCI+ were disadvantaged relative to the other diagnostic conditions. Finally, informant-rated caregiver burden declined at follow-up assessment in the CCT condition relative to the AC condition.	Increase in memory, Increase in global cognition, Increase in learning, Increase in mood	RCT
Kiosses D.N., et al. [41]	Older adults with major depression, cognitive impairment (up to the level of moderate dementia)	Two home-delivered psycho-social treatments, Problem Adaptation Therapy (PATH) and Supportive Therapy for Cognitively Impaired Older Adults (ST-CI)	Montgomery Asberg’s Depression Rating Scales’ (MADRS) Negative Emotions scores were significantly associated with suicidal ideation during treatment	Improved negative emotions, significantly predicted reduction in suicidal ideation at follow-up interview	RCT comparing the efficacy of PATH versus ST-CI
Soilemezi D., et al. [44]	Carers of people with dementia	Interview with carers in their home	No control group. Aspects of the architectural and interior environment (e.g., size, condition, layout and accessibility, familiarity) are perceived as important as well as a plethora of environmental strategies that encourage independence and comfort at home	All the carers recommended the use of equipment to increased quality of life for both the carer and the person with dementia	Cross-sectional qualitative study analyzing experience of caregivers
Me R., et al. [43]	Adult caregivers for dementia pts, 55% female	Survey on caregivers to assess acceptability, wearability, setting suitability, usability, and general concept about wearables for navigation for those with dementia.	No control group. Survey responses showed high usability and acceptability of device, but positioning is problematic, and it is unlikely it would be worn for very long.	Not reported	Qualitative survey concerning wearables for navigation
Teunissen L., et al. [45]	Adult patients with dementia, family member, a healthcare professional or a recreational therapist	eHealth CRDL, an interactive instrument	No control group. Family members and professionals reported that interaction with CRDL and dementia patients was a pleasant experience. In group interactions, the participants were fascinated by it, but needed active encouragement to interact	Not reported	Qualitative observational study, one on one integration, or group integration
Darragh M., et al. [35]	Adults with memory challenges, mild cognitive impairment, or mild dementia. Carers of those patients and those with expertise in MCI or dementia	Survey about a homecare robot	No control group. Robot can provide both therapeutic and practical benefit for mildly cognitive impaired patients. Provide reassurance, track health and well being	Improve health and wellbeing of dementia patients, monitor psychological wellbeing,	Semi-structured Interviews
Garzon-Maldonado G., et al. [38]	Adult caregivers of patients with dementia	Survey instrument asked about a telephone assistive intervention	No control group. 73.6% of the caregivers consider telephone assistive systems (TAS) a better or much better system than on-site assistance, while only 2.6% of the caregivers considered TAS a worse or much worse system than on-site assistance	Not reported	Mixed methods
Lazarou I., et al. [56]	Adults with dementia	Home monitoring	No control group. Interventions included wearable, sleep, object motion, presence, and utility usage sensors. Positive effects on movement intensity, cognitive function, activities of daily living (ADLs), and behavior are attributed to early detection of problems, objective measurements, and direct guidelines from provider to patient.	Improvement in movement intensity, Improved cognitive function, Improved ADLs, Improved behavior	Case study
Purves B.A., et al. [64]	Dementia patients	Computer Interactive Reminiscence Conversation Aid (CIRCA)	No control group. The computer serves as an additional participant in the conversation.	Computer-mediated interactions enable greater engagement and more enjoyment than a traditional method of story capture.	Focus groups and pilot Study
Newton L., et al. [57]	General practitioners, Patients with dementia, carers	Survey instrument inquired about AT from providers, caregivers, and those with dementia	No control group. This study raised awareness of AT just by asking about it, and it helped care givers find information on AT	Not reported	Qualitative methods with semi-structured interview and thematic analysis
Egan K.J. and Pot A.M. [52]	People with dementia and care givers	Focus group to discuss AT	No control group. Assistive health technology is early in development; however, its promise to improve the lives of those with dementia and their caregivers is great.	Not reported	Qualitative, exploratory, focus groups
Wolters M.K., et al. [58]	Adults with dementia, family caregivers, older people without diagnosis of dementia	Intelligent cognitive assistant	No control group. Participants varied in the technology they owned, the technology they used, their sense of proficiency, and their attitudes towards technology. The speaking calendar was helpful.	Improvement to cognitive decline	Questionnaire in focus group
Gibson G., et al. [61]	People with dementia, carers	Semi-structured interviews on everyday use of AT	No control group. Three categories of AT were identified as most common: those accessed through social care services, those purchased “off-the-shelf”, and “do-it-yourself”	Not reported	Semi-structured interview
Hattink B.J., et al. [53]	Dementia Care	Questionnaire on the intervention Rosetta, which was installed in the homes of patients.	All participants concluded that Rosetta is a very useful development in aiding dementia patients	Not reported	Controlled trials with pre- and post-test measures
Adolfsson P., et al. [48]	Twelve adults with cognitive disabilities and experience of using electronic planning devices (EPDs)	Electronic planning devices	No control group. Two themes emerged from the five categories: An increasing desire to participate and a need for necessary individual adaptation of the electronic daily planner (EPDs). The themes describe how the environment influences the use of EPDs.	Device enhanced their ability to concentrate and focus, resulting in less stress and forgetfulness	Interview and qualitative analysis
Czarnuch S., and Mihailidis A. [51]	Older adults with dementia	Intelligent AT, random-decision forest (computer learning)	No control group. Precision and recall were 0.994 and 0.938 for the depth tracker compared to 0.981 and 0.822 for the color tracker with the current data, and 0.989 and 0.466 in the previous study.	Not reported	Non-experimental
Jupiter T. [55]	Older adults with dementia	Hearing AT	No control group. No differences were found on the Mini Mental Status Evaluation (MMSE) pre- to post-fitting. Self and Staff Nursing Home Hearing Handicap Index (NHHI) scores were similar.	Improve hearing	Pilot study, longitudinal time-series
Boger J., et al. [50]	Two engineers, two computer scientists, a human factors expert, a rehabilitation scientist, a statistician, two clinical research assistants and 27 older adult participants ranging from cognitively intact to advanced stages of dementia	Self-reporting by asking participants to rate how difficult they found using the tap to be through a verbally administered, single four-point Likert scale question	No control group.	Not reported	Two sub-studies: (1) determining usability factors and evaluating tap usability and (2) automating the analysis of tap use
Jawaid S., and McCrindle R. [54]	Elderly adults	Computerized Help Information and Interaction Project (CHIIPS) program	No control group. Participants found CHIIPS to be useful, especially for people with age related memory loss or mild dementia	Not reported	Qualitative (questionnaire) collected opinions on the usefulness of the application
Fardoun H.M., et al. [37]	People with dementia	Analyzing pictures taken by a smart watch, which the patient carries; the person in front is recognized and information about him is sent to the watch.	No control group. Prototype showed correct results as a personal information system based on face recognition. However, usability flaws were identified in the interaction with the smart watch	Not reported	Qualitative, exploratory
Kerssens C., et al. [62]	Older adults and caregivers as dyads	The Companion (touch screen technology) that delivers non-drug, psychosocial intervention	No control group. Technology was easy to use, significantly facilitated meaningful and positive engagement, and simplified caregivers’ daily lives	Made their daily lives easier and made helping their spouse easier	Interview, implementation study
Arntzen C., et al. [49]	Young People (under 65) Living with Dementia	Incorporation of AT into everyday life for young people with dementia	No control group. AT are being negotiated and transformed by active embodied and situated social subjects in their everyday life	Not reported	Qualitative longitudinal study
Woodberry E., et al. [65]	Alzheimer’s patients	Wearable camera to improve memory	No control group. SenseCam outperformed the diary method for 5/6 people in the study	Help patients recall past events	Longitudinal design
Corbett A., et al. [60]	Eligible participants were older than the age of 50, and had access to a computer and the Internet	Compared evidence-based reasoning and problem-solving cognitive training (ReaCT), general cognitive training (GCT), and a control treatment	Online cognitive training (CT) confers significant benefit to cognition and function in older adults, with benefit favoring the Reasoning package. Scale of benefit is comparable with in-person training, indicating its potential as a public health intervention	Significant benefit to activities of daily living in a group of adults older than 60 receiving both the online GCT and ReaCT interventions compared with control, over a period of six months	Double-blind six-month online randomized controlled trial
Adolfsson P., et al. [59]	People with psychiatric or neuropsychiatric disorders (the respondents had to be ≥18 years old, had to have at least two months experience in using an EPD and have regular contact with a prescriber)	Interviewed with support from a study specific guide to explore the subjective experiences of people with cognitive disabilities in relation to the use of EPDs	No control group. EPDs can help people with cognitive disabilities not only in planning but also in organization and managing time	Use of EPDs seems to reduce the gap between the challenges the respondents encounter in everyday life and their cognitive disability; increased autonomy	Qualitative content analysis
Mao H., et al. [63]	The study consisted of caregivers perceptions of the usefulness of AT devices and Expert agreement on the common indicators of AT devices	Questionnaires to determine high perceived usefulness vs low perceived usefulness	No control group. The results suggested that caregivers preferred technologies that prevented accidents over technologies that informed them of the occurrence of an accident or that only managed the consequences.	There were no medical outcomes, the study was done to view the caregivers’ perception on the usefulness of AT devices	Qualitative (questionnaire)

**Table 2 healthcare-08-00278-t002:** Summary of evidence.

Authors	Intervention Theme	Medical Outcomes Themes	Facilitator Themes	Barrier Themes
Cruz-Sandoval D. and Favela J. [19]	Social robots	Increased talking	Not reported	Not reported
		Increased talking		
Thomas N.W.D., et al. [21]	Telehealth	No effect on depression	AT are invisible to users	Not reported
			No personal interface necessary	
Kenigsberg P.A., et al. [3]	Interview	Reduced sensory loss	Increased safety	Cost
		Improved coping	Increased wellbeing	Users forget how to use them
		Improved cognitive abilities	Increased independence	Complex interfaces
				Liability concerns
Czarnuch S., et al. [20]	Predictive model	Not reported	Not reported	Cost
Billis A., et al. [23]	Interview	Increased autonomy	Continuous training	Complex interfaces
				Users forget how to use them
				PWD do not want AT
				Ethical concerns
				Technical literacy
Lancioni G.E., et al. [27]	Mixed interventions	Increased ADLs	Not reported	Not reported
Lancioni G.E., et al. [28]	Walker with technology	Increased step frequency	Increased involvement	Not reported
Olsson A., et al. [30]	Sensors with voice	Increased autonomy	Increased safety	Complex interfaces
			Increased independence	Poor sound quality
			Increased self-confidence	Limitations of device
			Increased mindfulness	
Nauha L., et al. [29]	Interview	Not reported	Not reported	Development must improve capabilities
				Complex interfaces
				Poor sound quality
Holthe T., et al. [25]	Interview	Increased autonomy	Increased safety	Degenerative nature of AD complicates timing of AT used
		Decreased stress	Simple interfaces	Development must improve capabilities
			Visual reminders	Needs more government involvement
			Auditory reminders	Complex interfaces
Wilz G., et al. [33]	Telephone	Improved mood	Not reported	Not reported
		Improved depression		
		Improved overall health		
		Improved coping		
		Improved behavior		
Valdivieso B., et al. [32]	Telephone	Increased ADLs	Not reported	Not reported
Collins, M. [24]	Interview	Fewer medication errors	Continuous training	Degenerative nature of AD complicates timing of AT used
		Fewer falls	Remote monitoring	PWD do not want AT
		Improved behavior	Visual reminders	
Asghar I., et al. [22]	Interview	Improved overall health	Increased communication	PWD do not want AT
			Increased involvement	Font size
			Visual reminders	Limitations of device
Jouvelot P., et al. [26]	Cognitive stimulation	Increased autonomy	Visual reminders	Development must improve capabilities
			Remote monitoring	
Thoma-Lürken T., et al. [31]	Interview	Not reported	Not reported	Safety
Kouroupetroglou C., et al. [42]	Social robots	Not reported	Increased communication	Degenerative nature of AD complicates timing of AT used
				Development must improve capabilities
Jiancaro T., et al. [40]	Mixed interventions	Not reported	Increased independence	Degenerative nature of AD complicates timing of AT used
				Cost
Jarvis F., et al. [39]	Interview	Not reported	Simple interfaces	Lack of awareness of availability of AT
Dethlefs N., et al. [36]	Cognitive stimulation	Not reported	Caregivers want AT	Degenerative nature of AD complicates timing of AT used
Wang R.H., et al. [47]	Social robots	Not reported	Caregivers want AT	PWD do not want AT
				Could decrease interaction with caregiver
Toots A., et al. [46]	Walker with technology	Increased step frequency	Increased independence	AT could mask changes in gait over time
Bahar-Fuchs A., et al. [34]	eHealth	Improved memory	Not reported	Not reported
		Improved cognitive abilities		
		Improved mood		
		Improved memory		
Kiosses D.N., et al. [41]	Telehealth	Improved mood	Increased safety	Not reported
		Improved depression		
Soilemezi D., et al. [44]	Interview	Increased ADLs	Caregivers want AT	Not reported
Me R., et al. [43]	Interview	Not reported	Caregivers want AT	Development must improve capabilities
Teunissen L., et al. [45]	eHealth	Not reported	Increased communication	Degenerative nature of AD complicates timing of AT used
Darragh M., et al. [35]	Social robots	Improved overall health	Increased safety	Cost
Fardoun H.M., et al. [37]	Cognitive stimulation	Not reported	Increased wellbeing	Font size
Garzon-Maldonado G., et al. [38]	Telephone	Not reported	Cost benefits	Not reported
			Caregivers want AT	
Lazarou I., et al. [56]	Tele-monitoring	Improved cognitive abilities	Not reported	Not reported
		Improved behavior		
		Increased ADLs		
Newton L., et al. [57]	Interview	Not reported	Increased safety	Not reported
Egan K.J. and Pot A.M. [52]	Focus group	Not reported	Not reported	Users forget how to use them
				Needs more government involvement
				Cost
				Cost
				Degenerative nature of AD complicates timing of AT used
Wolters M.K., et al. [58]	Cognitive stimulation	Improved cognitive abilities	Auditory reminders	Not reported
Gibson G., et al. [61]	Interview	Not reported	Can be provided by government	Could decrease interaction with caregiver
				Cost
Hattink B.J., et al. [53]	Interview	Not reported	Increased independence	Development must improve capabilities
Adolfsson P., et al. [48]	Electronic planners	Decreased stress	Increased self-confidence	Development must improve capabilities
		Improved memory	Increased mindfulness	
Czarnuch S., and Mihailidis A. [51]	Cognitive stimulation	Not reported	Increased independence	Complex interfaces
Jupiter T. [55]	Hearing AT	Improved overall health	Increased mindfulness	PWD do not want AT
Boger J., et al. [50]	Interview	Not reported	Simple interfaces	Not reported
Jawaid S., and McCrindle R. [54]	Cognitive stimulation	Not reported	Cost benefits	Not reported
			Simple interfaces	
Arntzen C., et al. [49]	Interview	Not reported	Increased wellbeing	Not reported
Kerssens C., et al. [62]	Social robots	Improved coping	Increased wellbeing	PWD do not want AT
Purves B.A., et al. [64]	eHealth	Improved mood	Increased communication	Not reported
Woodberry E., et al. [65]	Cognitive stimulation	Improved memory	Increased self-confidence	PWD do not want AT
Corbett A., et al. [60]	Cognitive stimulation	Increased ADLs	Not reported	Not reported
		Improved cognitive abilities		
Adolfsson P., et al. [59]	Interview	Improved cognitive abilities	Increased independence	Not reported
		Increased autonomy	Increased wellbeing	
Mao H., et al. [63]	Interview	Not reported	Caregivers want AT	PWD do not want AT

**Table 3 healthcare-08-00278-t003:** Summary of quality assessments.

Strength of Evidence	Frequency of Occurrence	Quality of Evidence	Frequency of Occurrence
III (Non-experimental, qualitative)	31 (65%)	B (Good quality)	39 (81%)
I (Experimental study or RCT)	7 (15%)	A (High quality)	9 (19%)
II (Quasi-experimental)	5 (10%)	C (Low quality or major flaws)	0 (0%)
IV (Opinion)	5 (10%)		
a		b	

**Table 4 healthcare-08-00278-t004:** Affinity matrix of ATs.

Intervention	References	Occurrences *n* = (48)	Frequency
Interview	[3,22,23,24,25,29,31,39,43,44,49,57,59,61,63,65]	16	33%
Cognitive stimulation	[26,36,37,51,53,54,58,60,65]	9	19%
Social robots	[19,35,42,47,62]	5	10%
Telephone	[32,33,38]	3 *	6%
eHealth	[34,45,64]	3	6%
Walker with technology	[28,46]	2	4%
Telehealth	[21,41]	2	4%
Mixed interventions	[27,40]	2	4%
Tele-monitoring	[56]	1	2%
Predictive model	[20]	1	2%
Electronic planners	[48]	1	2%
Focus group	[52]	1	2%
Sensors with voice	[30]	1	2%
Hearing AT	[55]	1	2%

* Multiple occurrences in one study.

**Table 5 healthcare-08-00278-t005:** Affinity matrix of facilitators to the use of ATs.

Facilitators	References	Occurrences *n* = (68)	Frequency
Not reported	[19,20,27,29,31,32,33,34,52,56,60]	11	16%
Caregivers want AT	[36,38,39,43,44,47,53,63]	8	12%
Increased independence	[3,30,40,46,51,53,59]	7	10%
Increased safety	[3,25,30,35,41,57]	6	9%
Increased wellbeing	[3,37,49,59,62]	5	7%
Increased communication	[22,42,45,64]	4	6%
Simple interfaces	[25,39,50,54]	4	6%
Visual reminders	[22,24,25,26]	4	6%
Increased self-confidence	[30,48,65]	3	4%
Increased mindfulness	[30,48,55]	3	4%
Remote monitoring	[24,26]	2	3%
Continuous training	[23,24]	2	3%
Cost benefits	[38,54]	2	3%
Increased involvement	[22,28]	2	3%
Auditory reminders	[25,58]	2	3%
AT are invisible to users	[21]	1	1%
No personal interface necessary	[21]	1	1%
Can be provided by government	[61]	1	1%

**Table 6 healthcare-08-00278-t006:** Affinity matrix of barriers to the use of ATs.

Barriers	References	Occurrences *n* = (75)	Frequency
Not reported	[19,21,27,28,32,34,38,41,44,49,50,54,56,57,58,59,60,64]	19	25%
Cost	[3,20,35,39,40,52,61]	8 *	11%
PWD do not want AT	[22,23,24,47,55,62]	8 *	11%
Complex interfaces	[3,23,25,29,30,51,53]	7	9%
Development must improve capabilities	[25,26,29,42,43,48,53]	7	9%
Degenerative nature of AD complicates timing of AT used	[24,25,36,40,42,45,52]	7	9%
Users forget how to use them	[3,23,52]	3	4%
Could decrease interaction with caregiver	[47,61]	2	3%
Font size	[22,37]	2	3%
Poor sound quality	[29,30]	2	3%
Limitations of device	[22,30]	2	3%
Needs more government involvement	[25,52]	2	3%
AT could mask changes in gait over time	[46]	1	1%
Liability concerns	[3]	1	1%
Lack of awareness of availability of AT	[39]	1	1%
Safety	[31]	1	1%
Ethical concerns	[23]	1	1%
Technical literacy	[23]	1	1%

* Multiple occurrences in one study.

**Table 7 healthcare-08-00278-t007:** Affinity matrix of medical outcomes commensurate with the use of ATs.

Medical Outcomes	References	Occurrences *n* = (69)	Frequency
Not reported	[20,29,31,36,37,38,39,40,42,43,45,47,49,50,51,52,53,54,57,61,63]	23 *	33%
Improved cognitive abilities	[3,34,56,58,59,60]	6	9%
Increased ADLs	[27,32,44,56,60]	5	7%
Increased autonomy	[23,25,26,30,59]	5	7%
Improved memory	[34,48,65]	4 *	6%
Improved overall health	[22,33,35,55]	4	6%
Improved mood	[33,34,41,64]	4	6%
Improved behavior	[24,33,56]	3	4%
Improved coping	[3,33,62]	3	4%
Improved depression	[33,41]	2	3%
Increased step frequency	[28,46]	2	3%
Decreased stress	[25,48]	2	3%
Increased talking	[19]	2 *	3%
No effect on depression	[21]	1	1%
Reduced sensory loss	[3]	1	1%
Fewer medication errors	[24]	1	1%
Fewer falls	[24]	1	1%

* Multiple occurrences in one study.

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
