# Peer review of "Evaluating the Facilitators, Barriers, and Medical Outcomes Commensurate with the Use of Assistive Technology to Support People with Dementia: A Systematic Review Literature"

_healthcare, 2020, doi:10.3390/healthcare8030278_

Round 1

Reviewer 1 Report

The review of article healthcare-879876 under the title: Evaluating the facilitators, berries, and medical outcomes commensurate with the use of assistive technology to support people with dementia: a systematic review literature.     

This is an interesting and well-written article, and the English seems to be grammatically correct (nevertheless, I cannot be considered as English native). The authors analyzed the data published in the last 5 years, which described the usefulness of different assisted technologies in dementia treatment. It is really good article required only minor correction.

Lines 54, 183 – please remove the double square brackets.

Lines 81 and 88 – please don’t repeat the same information.

Number of table 3 is repeated. Please correct the order of the tables.

Author Response

Lines 54, 183 – please remove the double square brackets.

** Thank you for finding these. I found three instances of double brackets and corrected them.

Lines 81 and 88 – please don’t repeat the same information.

** I believe you are referring to the repetition of "5 years". I do not believe it is a repeat. The first time it is used, 5 years is listed as an eligibility criteria. The second time it is used, an explanation for 5 years is provided.  I moved the second instances of "5 years" to immediately follow the first so that it does not seem accidental.

Number of table 3 is repeated. Please correct the order of the tables.

** Thank you for catching this.  I corrected the error, and I changed the number listed on subsequent tables.

Reviewer 2 Report

This review is to identify the facilitators, barriers, and medical outcomes commensurate with the use of assistive technologies with people living with dementia. The systematic review revealed the positive relations that occur when people living with dementia and their caregivers use assistive technologies. Overall, the review is well-written with minor grammatical errors that do not necessarily detract from the clarity of the work. However, form the current version of this review there are some weaknesses that the authors should particularly pay attention and handle:

  1. The authors have compared their work with four studies from the year 2015 to 2018. why didn't they also compare with studies from 2019 or 2020?
  2. What are the differences between your study and these studies:
    1. Kang, H. S., Makimoto, K., Konno, R., & Koh, I. S. (2019). Review of outcome measures in PARO robot intervention studies for dementia care. Geriatric Nursing.
    2. Van Boekel, L. C., Wouters, E. J., Grimberg, B. M., van der Meer, N. J., & Luijkx, K. G. (2019). Perspectives of stakeholders on technology use in the care of community-living older adults with dementia: A systematic literature review. In Healthcare.
    3. Rondon-Sulbaran, J., Daly-Lynn, J., McCormack, B., Ryan, A., & Martin, S. (2019). An exploration of the experiences of informal carers supporting a relative living with dementia during and after the move to technology-enriched supported accommodation. Ageing & Society.
    4. Sriram, V., Jenkinson, C., & Peters, M. (2020). Carers’ experience of using assistive technology for dementia care at home: a qualitative study. BMJ Open.
  3. The authors mentioned that they have removed a lot of reviews and worked only on 48 studies. However, in Table1 they have listed 50 studies. So which is the correct?
  4. The abbreviation must be written beside the words when it appears for the first time. So, the authors need to check all of them. For example, "PWD" and "AT" is mentioned two times. The first is in the abstract section, while the second time is on page two. Also, "PROSPERO" was mentioned without giving any meaning.
  5. Also, references should be written between these ”[]” instead of these ”[[]]”, check page two and page four. Also, reference number two was written three times in the same paragraph, It would be better to write it only once.

Author Response

1.  We compared the results of our review with other similar reviews.  I assume the specifics of your comment for comment 1 are listed in comment 2.  I will address each item separately in comment 2.

2.  I have included items 1,2, 3, and 4 in the discussion. Our results were commensurate with these reviews.  

3.  Thank you for pointing this out. There were two duplicates in Table 1. I have removed them.

4.  With AT and PWD, I assume you want me to remove the second time I introduce the abbreviation (1st in abstract, 2nd in introduction). I removed the second introduction. PROSPERO is not an abbreviation.  Those who named the international database chose to use all caps to differentiate it from the protagonist of William Shakespeare's play.

5.  Thank you for pointing these out.  I found three instances of double brackets.  I corrected them.

Reviewer 3 Report

Dear authors,

thank you for your interesting review on facilitator, barriers and outcomes of 

I have a major concern about discussions.

Despite your findings show a dramatic lack of results in terms of outcome and several biases in the studies, you argue that "Health policy makers should consider augmenting AT development through grants or subsidies. As the aging of society progresses, so will the prevalence of PWD. Development of AT for PWD should be a priority. This corresponds with the theme [THAT?] needs more government involvement." Page 22, lines 296, and ss. This statement is not supported by your own findings. Please revise this paragraph, and add a discussion about the AT outcomes. At the beginning of your paper, the topic "AT outcomes" was presented as a key topic on which no previous revisions have been focused their attention. The readers would expect more on this point. I suggest discussing also the biases you found and how to read your results in light of them.

With respect to the results, it would be interesting to understand how outcomes were measured when PWDs were surveyed/interviewed: were structured and validated PROMs used? Are they comparable to each other? Can the eventually different measurement choices introduce biases in your results? I suggest adding this aspect to your analysis. Moreover, a discussion of this point should be also added.

Minor comments

Table page 13, last column: the study design of the papers of Jawaid & McCrindle, and Hattink and colleagues are differently defined (and detailed) in comparison to the others. The same for the paper of Mao, page 14. Please check the table and report the various info in a homogeneous manner.

Table page 15: why did you use bold for the first raw (study of Cruz-Sandoval and Favel)? I do not understand why you decide to repeat the themes in the Medical Outcome Themes column (i.e. first study: Increased talking / Increased talking). What information did the repetition in the frame add to the reader? Maybe it can be confusing. Could you please clarify this point?

Please, check the whole article for detecting typos. For instance, at Page 19 Line 215, a comma is missing between Netherland and Germany. Please, check this paragraph and make more clear what do you mean for "single organization" by listening then Countries. Are private or public organizations? are they multi-center studies (for this reason, there are some subsequent "and"?)?. 

Author Response

The reviewer did not number the comments, so I will do my best to address them all:

1. Despite your findings show a dramatic lack of results in terms of outcome and several biases in the studies, you argue that "Health policy makers should consider augmenting AT development through grants or subsidies.

** This comment was supported by the theme that further development must occur in the area of AT for PWD.  Development has, so far, been conducted with support by market demand, however, the results of development has been poor. That precipitated the suggestion for public funding. 

2. As the aging of society progresses, so will the prevalence of PWD.

** This is supported through national statistics. I provided a reference.

3.  Development of AT for PWD should be a priority. This corresponds with the theme [THAT?] needs more government involvement."

** I disagree with your suggestion. The phrase "needs more government involvement" is listed in italics because it is a theme.  This is how I have listed all themes throughout the manuscript.

4. Page 22, lines 296, and ss. This statement is not supported by your own findings. Please revise this paragraph, and add a discussion about the AT outcomes.

** I am confused by your reference.  I do not know what "ss" refers to.  Can you please revise your comment?  I am happy to look at the issue and consider the comment.  A paragraph is already provided for medical outcomes commensurate with the use of AT.

5. At the beginning of your paper, the topic "AT outcomes" was presented as a key topic on which no previous revisions have been focused their attention. The readers would expect more on this point.

** I am sorry. I find no reference to the quoted term "AT outcomes" in the manuscript.  There are no previous revisions to this manuscript (this is the first review).

6. I suggest discussing also the biases you found and how to read your results in light of them.

** A paragraph is already provided (page 20) on the topic of bias and the limitations of this bias to the external validity of results.

7.  With respect to the results, it would be interesting to understand how outcomes were measured when PWDs were surveyed/interviewed: were structured and validated PROMs used? Are they comparable to each other? Can the eventually different measurement choices introduce biases in your results? I suggest adding this aspect to your analysis. Moreover, a discussion of this point should be also added.

** A list of study design is provided in PICOS.  I am sorry, but I do not know what "PROM" refers to.  I assume these are outcome measures, but I need the full reference so I can fully understand your comment.  Please clarify.

Minor comments

8.  Table page 13, last column: the study design of the papers of Jawaid & McCrindle, and Hattink and colleagues are differently defined (and detailed) in comparison to the others. The same for the paper of Mao, page 14. Please check the table and report the various info in a homogeneous manner.

** I corrected these to make them more homogeneous.

9.  Table page 15: why did you use bold for the first raw (study of Cruz-Sandoval and Favel)?

** I removed the bold from Table 1.

10. I do not understand why you decide to repeat the themes in the Medical Outcome Themes column (i.e. first study: Increased talking / Increased talking). What information did the repetition in the frame add to the reader? Maybe it can be confusing. Could you please clarify this point?

** I specifically addressed this repetition in paragraph 3.4 before the table.  I define the differences and how they fell under the same theme but the observations are distinct (which can be found in the appendix).

11.  Please, check the whole article for detecting typos. For instance, at Page 19 Line 215, a comma is missing between Netherland and Germany.

** This is not an error: It is a series of countries in the same study.  You will find this similar combinations in each of the explanations.

12. Please, check this paragraph and make more clear what do you mean for "single organization" by listening then Countries. Are private or public organizations? are they multi-center studies (for this reason, there are some subsequent "and"?)?.

** I listed "single originations" not "single organizations." A single origination was a theme or observations listed from only one origin.

Reviewer 4 Report

The present work brings a review regarding to the evaluating of the facilitators, barriers and medical outcomes commensurate  with the use of AT on Dementia.

L6: This work did present from which University/Institution and Country they come from. Please, add it.

L17-L21: The sentence "This review" was written three times in a short period. Can you correct this?

L54: Double "[[". Please, correct it.

L63:"However, NONE of these....". Please, correct it.

L68: "..adoption [[,]] and medical..." Please, remove the comma (","). If possible, also in the Title.

L74: It is important to describe in few words, the Kruse protocol [ref 13]..."This review followed the Kruse protocol, WHICH it says/defines....".

L82: Too long space between "technology." and "A quality..."

L101-102: "..on academic or peer reviewed journals over the last 5 years.." . 5 years was the period of papers reviewed, is that? If yes, it is important to be shortly placed in the abstract and introduction.

L179: Please, it is better to replace the word "piloted form" by "applied form" or something similar. The term "piloted form" is not usual.

L185: Tables 1, 2, presented detailed and relevant information. But in Table 1, it is better to add the reference numbers for each Author, like as in Table 2.

L212: Table 3. Please remove the black columns in the middle. It brings nothing. Also, what kind of frequency the Table  3 is talking about? Please, clear these information.

L231: Tha tag "Table 3" was repeated instead of "Table 4". 

L249: "Table 4" must to be placed in the line 231. Here the correct it Table 5.

L272, 293: Correct the tag of the Tables to "Table 6" and "Table 7". (See the mistake in the line L212). 

L342: A bit confuse sentence. In the Section 2 (lines L101-102), was said "five years", and now you say "..six years"??

L345: The comma is not necessary here. "technologies[[,]] and this". Please, remove it.

L369: The conclusion is so short. It must to present in resume, the findings and possible improvement that were observed. Please, improve the conclusion.

L412: The appendix must to describe better the Tables that they present. It is a little confuse, to suddely jump to the Appendixes and see a lot Tables. If possible, please, put some description about each Appendix's Tables.

--------------------------------------------

In resume, the present paper brings a detailed review on the proposed context and after some minor reviews I just proposed, it can be accepted for me.

Author Response

L6: This work did present from which University/Institution and Country they come from. Please, add it.

** I added an affiliation line.

L17-L21: The sentence "This review" was written three times in a short period. Can you correct this?

** I used a pronoun to decrease the use of "This review"

L54: Double "[[". Please, correct it.

** Corrected

L63:"However, NONE of these....". Please, correct it.

** Corrected

L68: "..adoption [[,]] and medical..." Please, remove the comma (","). If possible, also in the Title.

** Corrected

L74: It is important to describe in few words, the Kruse protocol [ref 13]..."This review followed the Kruse protocol, WHICH it says/defines....".

** I added some description.

L82: Too long space between "technology." and "A quality..."

** I did not find the specific reference you mention.  I attempted to use two spaces after each end-of-sentence punctuation, however paragraphs are fully justified, which can give the appearance of additional spacing.

L101-102: "..on academic or peer reviewed journals over the last 5 years.." . 5 years was the period of papers reviewed, is that? If yes, it is important to be shortly placed in the abstract and introduction.

** This information is already listed under the methods paragraph of the abstract.

L179: Please, it is better to replace the word "piloted form" by "applied form" or something similar. The term "piloted form" is not usual.

** I replaced all four instances of "piloted form" with your preferred term.

L185: Tables 1, 2, presented detailed and relevant information. But in Table 1, it is better to add the reference numbers for each Author, like as in Table 2.

** Good point. I added references to Table 1.

L212: Table 3. Please remove the black columns in the middle. It brings nothing. Also, what kind of frequency the Table 3 is talking about? Please, clear these information.

** I removed the black bar. This was to distinguish between panels a and b.  I also added "of occurrence" after "frequency".

L231: Tha tag "Table 3" was repeated instead of "Table 4".

** This has been corrected.  Thank you.

L249: "Table 4" must to be placed in the line 231. Here the correct it Table 5.

** I believe this comment relates to the previous. Because Table 3 was listed twice, all subsequent tables had to be renamed.

L272, 293: Correct the tag of the Tables to "Table 6" and "Table 7". (See the mistake in the line L212).

** This has been corrected.

L342: A bit confuse sentence. In the Section 2 (lines L101-102), was said "five years", and now you say "..six years"??

** The review covered articles published in the last five years.  The "six years" refers to reviews that have published on a similar topic. They are not related.

L345: The comma is not necessary here. "technologies[[,]] and this". Please, remove it.

** I disagree. The phrase following the comma is an independent clause, and therefore, it requires the use of a comma.

L369: The conclusion is so short. It must to present in resume, the findings and possible improvement that were observed. Please, improve the conclusion.

** I improved the conclusion section.

L412: The appendix must to describe better the Tables that they present. It is a little confuse, to suddely jump to the Appendixes and see a lot Tables. If possible, please, put some description about each Appendix's Tables.

** The appendices are introduced in the results section. Each appendix is described through a detailed description after the appendix number.

Round 2

Reviewer 2 Report

This review is to identify the facilitators, barriers, and medical outcomes commensurate with the use of assistive technologies with people living with dementia. The systematic review revealed the positive relations that occur when people living with dementia and their caregivers use assistive technologies. However, the authors should particularly pay attention and handle:

  1. At the end of page 1 and the beginning of page 2, reference number [2] mentioned three times in the same paragraph so, It would be better to write it only once. Also, reference number [3] in the next paragraph has the same problem.
  2. The authors should check that all the abbreviations. They have to write them beside the word if they appear for the first time then, use these abbreviations instead of the original words. For example, assistive technologies (AT) mentioned at the line "14" so, at the line "57" the authors should use AT not assistive technologies again

Author Response

  1. I deleted subsequent uses of reference 2 and 3.  I do not agree with this recommendation because these are declarative statements, and as such, they require substantiation.
  2. I was able to find a few other abbreviations that had not been spelled out the first time. I found and corrected all uses of AT that were spelled out and abbreviated them, except where it was the beginning of a sentence.  I corrected these.